# Thyroxine and Thyroid-Stimulating Hormone in Own Mother’s Milk, Donor Milk, and Infant Formula

**DOI:** 10.3390/life12040584

**Published:** 2022-04-14

**Authors:** Réka A. Vass, Gabriella Kiss, Edward F. Bell, Attila Miseta, József Bódis, Simone Funke, Szilvia Bokor, Dénes Molnár, Balázs Kósa, Anna A. Kiss, Timea Takács, Flóra Dombai, Tibor Ertl

**Affiliations:** 1Department of Obstetrics and Gynecology, University of Pécs Medical School, 7624 Pecs, Hungary; bodis.jozsef@pte.hu (J.B.); funke.simone@pte.hu (S.F.); kiss.anna.aranka@gmail.com (A.A.K.); takacs.timea@pte.hu (T.T.); dombai.flora@pte.hu (F.D.); ertl.tibor@pte.hu (T.E.); 2National Laboratory for Human Reproduction, University of Pécs, 7624 Pecs, Hungary; bokor.szilvia@pte.hu (S.B.); molnar.denes@pte.hu (D.M.); 3Department of Laboratory Medicine, University of Pécs Medical School, 7624 Pecs, Hungary; kiss.gabriella2@pte.hu (G.K.); miseta.attila@pte.hu (A.M.); 4Stead Family Department of Pediatrics, University of Iowa, Iowa City, IA 52242, USA; edward-bell@uiowa.edu; 5Department of Pediatrics, University of Pécs Medical School, 7624 Pecs, Hungary; 6Department of Interior, Applied and Creative Design, University of Pécs, 7624 Pecs, Hungary; kosa.balazs@mik.pte.hu

**Keywords:** preterm milk, term milk, thyroxine, TSH, total protein, hind milk, breastfeeding, breast milk, infant feeding, formula feeding

## Abstract

Breastfeeding is widely supported in clinical and home practices, and it is known that different forms of infant feeding differ in hormone content. Thyroid hormones have essential physiological roles. In our study, we examined thyroid-stimulating hormone (TSH), thyroxine, and albumin levels in breast milk produced for term (n = 16) or preterm (n = 15) infants throughout the first 6 months of lactation. Moreover, we analyzed these components in donor human milk and in three different infant formulas. Term and preterm breast milk samples were collected monthly. The two groups had similar levels of TSH (18.4 ± 1.4 vs. 24.7 ± 2.9 nU/L), but term milk contained higher amounts of thyroxine (11,245.5 ± 73.8 vs. 671.6 ± 61.2 nmol/L) during the examination period. The albumin level was significantly higher in preterm breast milk than in term breast milk (328.6 ± 17.1 vs. 264.2 ± 6.8 mg/L). In preterm breast milk we detected downward trends in the levels of TSH (−30.2%) and thyroxine (−29.2%) in the 3rd through 6th month compared to the first 2 months of lactation. Microbiological safety of donor milk was ensured by Holder pasteurization (HoP). From the Breast Milk Collection Center of Pécs, Hungary, we enrolled 44 donor mothers into the study. HoP decreased TSH (−73.8%), thyroxine (−22.4%), and albumin (−20.9%) concentrations. Infant formulas used by the Neonatal Intensive Care Unit of the University of Pécs were found to not contain the investigated hormones, but their albumin levels were similar to the breast milk samples. The present study shows the lack of thyroid hormones in infant formulas compared to human milk and raises the question of whether formula-fed infants should be supplemented with thyroid hormones.

## 1. Introduction

Transplacental hormonal exposure impacts intrauterine development. Premature birth disrupts the placentofetal connection. Therefore, for a preterm infant, after birth breast milk is the absolute best source of maternal hormones. Breast milk is the recommended form of infant feeding, supported by numerous organizations worldwide. Adequate nutrition of preterm infants remains a challenge with regard to achieving a sufficient amount of bioactive substances, assuring linear growth and neurocognitive development [1,2]. During fetal life, hormones are available from the maternal circulation and amniotic fluid until the development of the fetal endocrine system is completed. For feeding preterm infants, the infant’s own mother’s milk is the first choice. If this is not available, donor milk (DM) is recommended [3]. It is important to note that the breast milk composition of mothers who gave birth to term infants differs from maternal milk produced for preterm infants. It was also proven that human milk reduces the incidence of necrotizing enterocolitis and several morbidities compared to formula-fed infants. For microbiological safety, DM is exposed to Holder pasteurization (HoP). However, this procedure influences the levels of bioactive factors and compounds in human milk [4,5].

During fetal life, the thyroid gland secretes thyroxine (T4) and triiodothyronine into the circulation from about 12 weeks’ gestation, and its levels increase until term. In limited amounts, the maternal T4 crosses the placenta throughout gestation, and this plays a critical role in the development of the central nervous system in the first trimester, as demonstrated by the fetal neurological impairment described in fetomaternal Pit-1 deficiency [6] and serious iodine deficiency [7]. From midgestation, the expression of the thyrotrophin releasing hormone (TRH) from the hypothalamus, the production of the thyroid-stimulating hormone (TSH) the pituitary gland, and the thyroidal expression of T4 rose steadily until the 36th gestational week. Physiological maternal thyroid function may play an important role in ideal neurological development even when the fetal thyroid gland turns to an autonomous organ [8]. Enzymatic deiodination adjusts the bioactivity of thyroid hormones in peripheral tissues [9]. Hypothyroidism is frequently observable among preterm infants due to the immaturity of the hypothalamic-pituitary-thyroid axis. Most such cases are related to delayed TSH elevation [10]. Hypothyroidism causes many complications, e.g., neurocognitive delay or cholestasis [11]. Thyroid functions influence intrauterine development and postnatal adaptation, e.g., maternal TRH treatment, which improves lung development [12]. Thyroid hormones are present not only in maternal and fetal plasma, but also in the amniotic fluid [13]. Mainly, albumin is a known carrier protein for different compounds, such as steroids or fatty acids, and thyroid hormones in the circulation and plays a crucial role in stabilizing extracellular fluid volume by participating in the regulation of the oncotic pressure (also known as colloid osmotic pressure) of plasma [14].

In the present study, we collected breast milk samples from mothers who delivered preterm or term infants, and measured the thyroxine, TSH, and albumin levels in the samples. We measured these substances in samples of donor milk and also attempted to detect the presence of the hormones and albumin in infant formula. The purpose of our investigation was to provide more information about the hormonal content of own mother’s milk, donor milk, and commercial infant formula.

## 2. Materials and Methods

The present study was conducted after the approval of the Regional and Local Research Ethics Committee of the University of Pécs, Pécs, Hungary (PTE KK 7072-2018). Waivers for participant consent were obtained.

The subjects were mothers who had given birth to term (n = 16) or preterm (n = 15) infants at the Department of Obstetrics and Gynecology, University of Pécs. Breast milk sample collection started at the 4th postpartum week and continued monthly until six months after birth. Participants pumped their entire breast expression into sterile polypropylene bottles between 1 p.m. and 3 p.m., and 5 mL from the bottle was poured separately and then stored in a sterile polypropylene tube until analysis.

As a part of our study, 44 mothers—all of them registered and approved donors—were involved from the Breast Milk Collection Center of the Unified Health Institution at Pécs, Hungary. Based on the breast milk collection centers’ protocol, mothers donated freshly pumped milk. We collected breast milk samples at six random occasions in the milk bank. We took samples individually and then the milk samples were pooled and Holder pasteurized (30 min at 62.5 °C) at the laboratory of the Unified Health Institution (Pécs, Hungary). After the procedure, we took three samples from the pooled and Holder pasteurized donor milk and stored them at −80 °C until laboratory measurements. During the analysis, first, the samples were sonicated in order to disrupt the milk fat globule membranes and, as previously described [15], centrifuged at 15,000× *g* for 15 min. Then the skim milk was transferred to polypropylene tubes for measurements based on previous laboratory preparation methods [16].

Three different infant formulas were tested in our analysis to obtain information about the hormone composition of these fluids. Infant formulas were prepared and applied regularly in the Department of Neonatology, University of Pécs and were tested at five different time points: Nutricia Milumil Pepti Pronutra (Danone, Paris, France), Beba Optipro Hypoallergenic (HA) Start (Nestlé, Vevey, Vaud, Switzerland), and Beba Optipro HA Pre (Nestlé, Vevey, Vaud, Switzerland). Samples were taken at the Neonatal Intensive Care Unit ten times in the morning between 8 and 9 a.m. and were stored similarly to the breast milk samples, in sterile polypropylene tubes at −80 °C. An infant has three feeding options. The first is own mother’s milk; when it is not available, donor milk is the recommended form of infant feeding; the third possibility is infant formula. Figure 1 shows the experimental design of the present project.

The TSH assay employs monoclonal antibodies (Elecsys, Roche Diagnostics, Rotkreuz, Switzerland) specifically directed against human TSH. For TSH detection, first, a 50 μL breast milk sample was mixed with a biotinylated monoclonal TSH-specific antibody and a monoclonal TSH-specific antibody labeled with ruthenium complex. Streptavidin-coated microparticles were added. The TSH detection range was 0.0005–100 mU/L. No crossreactivities were found with LH, FSH, human growth hormone, or human chorionic gonadotropin.

To measure thyroxine, a 15 μL breast milk sample and a thyroxine-specific antibody labeled with a ruthenium complex were mixed, and then biotinylated thyroxine and streptavidin-coated microparticles were added to the mixture. The thyroxine detection range was 0.3–100 pmol/L. The following cross-reactivities were found with L-triiodothyronine: 0.005%, D-triiodothyronine: 0.001%, reverse triiodothyronine: 0.003%, 3,3′,5-triiodothyroacetic acid: 0.0002%, and 3,3′,5,5′-tetraiodothyroacetic acid: 0.001%.

ARCHITECT i1000 system was applied for the measurements, and we followed the manufacturer’s instructions throughout. The fully automatized Cobas e 411 analyzer system performed all the measurements (Roche Diagnostics, Rotkreuz, Switzerland). After the previously described automatized preparation steps, the induced chemiluminescent emission was detected by a photomultiplier. Results were given after a two-point calibration curve was generated. Parallel with the breast milk samples, quality controls were tested.

Albumin was detected with an immunoturbidimetric assay (Roche Diagnostics, Rotkreuz, Switzerland). In this method, antialbumin antibodies react with the sample’s antigen and form complexes. After agglutination, the samples were measured turbidimetrically in a fully automatized way, and quality controls were also applied.

GraphPad (La Jolla, CA, USA) was used for statistical analysis, and Shapiro-Wilks tests were performed to test the data normality. Paired *t*-test or ANOVA was used for analysis. With post-hoc Dunnett test, repeated measures one-way ANOVA test was used to compare the effect of Holder pasteurization on the concentration of the chosen hormones. Differences were determined statistically significant when *p* values were <0.05. The study was powered to detect moderate effect sizes (Cohen’s d = 0.6). Results are presented as mean ± SEM or median values with interquartile ranges.

## 3. Results

### 3.1. Maternal Characteristics

All mothers involved in our study conceived spontaneously. Differences between preterm and term groups based on maternal age, body mass index (BMI), or infant sex were not found (Table 1). Mothers who were regular breast milk donors were more likely to have given birth to male infants. The recruited mothers were all of White race. None had chronic health conditions, e.g., diabetes previously or throughout their pregnancy or lactation period, and none of them followed any special diet. The age at donation for the donor mothers was 148.6 ± 12.8 days.

### 3.2. Preterm Breast Milk

Throughout the first 2 months of lactation, breast milk contained significantly higher amounts of thyroxine and TSH compared to the period between the third and sixth months. The albumin level of breast milk produced for preterm infants was similar across the first six months of lactation (Table 2).

### 3.3. Comparison of Preterm and Term Breast Milk

The concentrations of TSH, thyroxine, and albumin in breast milk produced for term infants did not change with time since birth. Therefore, we presented the average concentration values pooled from the first six month of lactation. While comparing the average thyroxine and TSH levels of preterm and term breast milk, thyroxine was present at significantly higher levels in term milk. In contrast, albumin was present at significantly higher levels in preterm milk, whereas the TSH level was the same in preterm and term breast milk (Table 3).

### 3.4. Effect of Holder Pasteurization on Breast Milk

TSH, thyroxine, and albumin levels were significantly decreased by Holder pasteurization. The TSH level was decreased with 73.8%, the thyroxine concentration with 22.4%, and the albumin level with an average 20.9% concentration (Table 4).

### 3.5. Infant Formula Analysis

Three of the infant formulas in use at the Department of Neonatology, University of Pécs are the Milumil Pepti Pronutra, the Beba Optipro HA Start, and the Beba Optipro HA Pre. Thyroxine and TSH were not found above the lower limits of detection in any of these formulas. Analyzing pooled results of the breast milk samples collected during the first 6 months of lactation, formulas had an average 315.58 ± 19.11 mg/L albumin level, which was similar to the levels measured in term and preterm breast milk.

## 4. Discussion

In the present study, we detected the presence of TSH, thyroxine, and albumin during the first six months of lactation in breast milk produced for term or preterm infants. The same components were analyzed in donor breast milk, and in commercial infant formulas as well, to provide previously unavailable information about the TSH, thyroxine, and albumin concentrations of three different feeding options of an infant.

Hormones and nutrients, including growth factors, immunoglobulins, chemokines, cytokines, oligosaccharides, vitamins, and immune cells have been demonstrated to transfer from the mother to the infant via breast milk [1,17]. The increasing knowledge of the important relationship of breastfeeding to infant outcomes highlights the value of breastfeeding. Thyroid hormones control the metabolism in almost every tissue and are essential for normal growth. Without the benefit of thyroid hormones, several tissues and organs, including the cells of the nervous system do not develop and function properly. In infants born before 30 weeks’ gestational age, the T4 and TSH surges are often lacking [18,19]. Therefore, premature neonates may experience ephemeral hypothyroxinemia interruption of the maternal thyroid hormone supply cutoff and immaturity of the preterm thyroid axis [20,21].

Regarding our knowledge about the presence of TSH in infant feeding sources, we concluded that TSH is measurable during the first 6 months of lactation in both preterm and term breast milk, but it is unavailable in infant formulas. The oral bioavailability of thyroxine is known, but it is not known whether TSH is bioavailable if provided orally. The presence of TSH receptors in the gastrointestinal tract raises the possibility of local effects [22].

In maternal milk produced for preterm infants, composition changes were detected. In the first two months of lactation, mother’s milk contains higher thyroxine and TSH compared to the time period between the 3rd and 6th month of breastfeeding. We hypothesize that after the suddenly interrupted pregnancy, maternal thyroid hormones may decrease slowly, with a drop present in the composition of breast milk. Sack et al. described a thyroxine peak in the 5th postpartum week at 4.3 μg/100 mL and concluded that the mother secretes 40–50 μg of thyroxine per day in her milk, covering the thyroxine need for an infant to avoid hypothyroidism [23].

Severe hypothyroidism that is often diagnosed in neonates proposes the possibility of the transplacental supply of thyroid hormones. Biological half-life of serum T4 was 3.6 days (between 2.7 and 5.3 days). In infants with thyroid agenesis and in neonates with a total organification defect, the serum T4 concentration and its disappearance kinetics were the same, suggesting that thyroxin also had a maternal origin in these infants. The authors concluded that in infants diagnosed with severe congenital hypothyroidism, a substantial amount of thyroxine is transferred from the mother to the fetus [24].

Gastric and intestinal digestion of proteins and other components are attenuated during the first few postnatal months. The infant’s gastrointestinal tract has immature barrier functions and the cell–cell connections. The amount of iodine through deiodination of T4 is insignificant in comparison with the inorganic iodine concentration of breast milk produced for preterm infants; the median concentration value was 105 μg/L. Iodine concentration showed a relatively large variation (38–293 μg/L) in breast milk, which might be caused by the fluctuating maternal iodine intake [25]. The gastrointestinal tract is having an active interaction with the T4, which is absorbing in considerably, but incompletely from the gut [22]. Via the blood–brain barrier or indirectly through the blood–CSF barrier, the thyroid hormone enters the brain [26].

In extremely preterm infants, transient low levels of plasma T4 is often detected and observed, described as the transient hypothyroxinaemia of prematurity. Van Wassenaer et al. investigated whether maternal milk is a relevant resource of thyroid hormones for very preterm infants and can relieve transient hypothyroxinemia [27]. Both the influence of breastfeeding on plasma thyroid hormone concentration and the thyroid hormone levels in preterm milk were analyzed. In a previous work using radioimmunoassay technique, the human milk thyroxine concentration had a mean value of 0.83 μg/L, resulting in a maximum T4 supply of 0.3 μg/kg via ingested breast milk. In their work, they compared the plasma thyroid concentration of breast milk-fed and formula-fed infants and no differences were found; therefore, the authors concluded that the amount of T4 present in breast milk and formula is too low to influence the hypothyroxinemic state of preterm neonates [27]. Many investigators have aimed to determine thyroid hormone content of human milk by applying various methods of sample preparation and determination. Previously, various average thyroid hormone levels have been described (T4, 0–77 μg/L) [23,28,29]. Assumably, this variability is caused by multiple factors, one of the most notable is the immature hypothalamo–pituitary–thyroid axis or the swift disruption of the maternal hormone pool and fetal thyroid hormone pools [24]. This hypothyroxinemic state can be filtered out by administering a daily T4 dose of 8 μg/kg [27]. Based on our results, a preterm infant receiving his or her own mother’s milk receives 5.2 μg of thyroxine per 100 mL, whereas term infants receive 8.7 μg/100 mL On the other hand, donor milk contains 3.9 μg thyroxine/100 mL. With average maternal milk thyroxine content, a preterm infant would need to be fed 154 mL/kg/d of mother’s milk to receive 8 μg/kg of thyroxine. In comparison, virtually no thyroxine is received by an infant fed only infant formula.

The reason for postnatal hypothyroxinemia is multifactorial, including the loss of the transplacental maternal T4 contribution, immaturity of the hypothalamic–pituitary axis, the responsiveness of the thyroid gland to TSH, and immaturity of peripheral tissue deiodination. In the first few weeks after birth, iodine balance is negative in very low birthweight babies, suggesting an inability to augment thyroidal iodine uptake and increase T4 secretion [27]. Previous studies have also shown that breast milk can provide an important and significant exogenous source of thyroxine to the infant [23]. Detection of thyroxine and TSH in breast milk has started in the radioimmunoassay detection era. The therapeutic dose of thyroxine is between 40–50 μg per day. Based on the average concentration values in our study, term infants received 10–16 times higher amounts of thyroxine compared to preterm infants, whereas TSH exposure was approximately the same between the preterm and term groups.

The effect of thyroxine supplementation was investigated previously in the studies of Amato et al. [30] and van Wassenaer et al., in which after thyroxine replacement, the Bayley Mental Development Index in the thyroxine-treated group was significantly higher compared to infants born at the same gestational age of 25–26 weeks [27]. In a randomized, double-blind trial, 153 infants born below 28 weeks of gestation were enrolled and supplemented with levothyroxine at 8 μg/kg/day or a placebo [31]. At the 42nd month of age, the neurodevelopmental outcome was tested by using Bayley III Mental and Psychomotor Developmental Indices. The levothyroxine-supplemented group had better motor, language, and cognitive function results. These data suggest that levothyroxine supplementation may improve the neurodevelopmental outcome. Case reports described the effectiveness and safety of intravenous levothyroxine administration in providing normalization of free T4 and TSH in infants with malabsorption after necrotizing enterocolitis [32]. Thyroid hormones involved in the development of fetal lung and maternal TRH treatment potentiate the corticosteroid effect in ameliorating neonatal respiratory distress syndrome [12]. Breast milk has a known protective effect against bronchopulmonary dysplasia (BPD). During the first 4 weeks of life, breastfeeding was associated with lower incidence of BPD as well as necrotizing enterocolitis, and extrauterine growth restriction in very low birth weight infants [33]. These observations highlight the importance of clinical investigations and studies focusing on early supplementation, especially in donor milk breast-fed or formula-fed preterm infants.

When own mother’s milk is not available in neonatal care, the recommended form of infant feeding is donor milk. Before feeding it to neonates, donor milk is pasteurized to ensure microbiological safety in human milk banks. The impact of this procedure on the concentration of various hormones is different. Pasteurization is necessary to assure the safety of donor milk, although it may affect milk quality by the reduction of bioactive components [4,6,15,34]. Our previous study detected TSH in similar amounts in breast milk produced for preterm and term infants, and Holder pasteurization increased the TSH level by 17%. When donor human milk was analyzed after HoP and 24-h storage in a refrigerator, the TSH content of donor milk was not affected [16]. In contrast, our present results showed a significant concentration decrease after HoP. Based on our result, TSH, thyroxine, and albumin levels were decreased by the procedure. To our knowledge, we detected for the first time decreases in thyroxine and albumin levels after HoP.

Albumin is a circulatory protein, with numerous vital physiological functions, such as regulating metabolic and vascular functions, maintaining microvascular integrity and oncotic pressure, providing antioxidant activities, binding ligands for different substances, and having anticoagulant effects. Its presence in human milk was known [35], but no previous studies examined the presence of albumin in breast milk produced for preterm infants, or investigated how HoP affects its level in breast milk. It has the indispensable role of providing a balanced complement of essential amino acids to the infant.

Our study had limitations. Sonication disrupts milk fat globules and allows proteins to enter the aqueous phase of the milk samples, but the removal of this fat layer during analysis may have led to an underestimation of hormone levels. Hormones were not measured in the blood of the infants, thus the impact of human milk content remains unknown. We investigated one method, the HoP; however, other techniques used to ensure microbiological safety of donor milk may differently influence the concentration of these breast milk components.

## 5. Conclusions

Our study highlights that different forms of infant feeding provide different amounts of thyroxine and TSH to the infant. Intrauterine development is influenced by numerous unreplaceable factors. After birth, bioactive compounds, such as hormones, impact postnatal development via absorption from the gastrointestinal tract. Preterm infants are born deficient in these transplacental hormones because their intrauterine development is interrupted early. Our results showed for the first time the long-term presence of TSH, thyroxine, and albumin in preterm and term breast milk during the first 6 months of lactation. The present study improves our knowledge of the composition of infant formulas by demonstrating the absence of measurable amounts of thyroxine and TSH in the formulas. Term infants receive higher thyroxine concentrations in their mothers’ milk than do preterm infants. It is known that HoP influences the composition of human milk. Our study showed that the HoP decreased the thyroxine content of breast milk. Oral thyroxine replacement is an accepted medical treatment. Its use should be investigated for exclusively formula-fed infants.

## Figures and Tables

**Figure 1 life-12-00584-f001:**
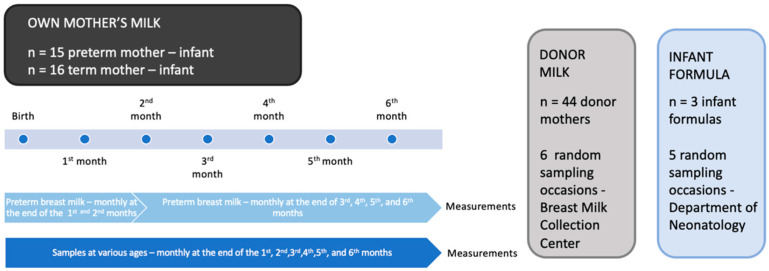
Experimental design of sample collection.

**Table 1 life-12-00584-t001:** Maternal characteristics.

	Preterm Maternal	Term Maternal	Donor
Number	15	16	44
Maternal age (years)	31.7 ± 1.1	32.1 ± 2.7	32.4 ± 0.5
Gestational age (weeks)	31.4 ± 2.1	39.6 ± 0.5	39.5 ± 0.2
Maternal BMI	27.8 ± 0.2	26.9 ± 0.5	26.3 ± 1.9
Gender of newborn			
Female	7	9	19
Male	8	7	25
Delivery			
Natural	4	11	27
Cesarean section	11	5	17

**Table 2 life-12-00584-t002:** Hormones in preterm breast milk.

Analyte	1st and 2nd Months	3rd–6th Months	*p*-Value
TSH, nU/L	23.2 ± 2.2	16.2 ± 1.8	0.0335
Thyroxine, nmol/L	842.2 ± 158.8	595.7 ± 49.2	0.0486
Albumin, mg/L	349.9 ± 34.1	318.3 ± 19.4	0.3919

**Table 3 life-12-00584-t003:** Albumin, thyroxine, and TSH content of preterm and term breast milk.

Analyte	Preterm (n = 90)	Term (n = 96)	*p*-Value
TSH, nU/L	18.4 ± 1.4	24.7 ± 2.8	0.0959
Thyroxine, nmol/L	671.6 ± 61.2	11,245.5 ± 73.8	<0.0001
Albumin, mg/L	328.6 ± 17.1	264.2 ± 6.8	0.0041

**Table 4 life-12-00584-t004:** Impact of Holder pasteurization on the concentrations of total protein thyroxine and TSH in donor milk (n = 44).

Analyte	Raw	HoP	*p*-Value
TSH, nU/L	20.6 ± 3.3	5.4 ± 0.6	<0.0001
Thyroxine, nmol/L	640.1 ± 32.4	506.1 ± 11.2	0.0072
Albumin, mg/L	289.1 ± 4.6	224.1 ± 5.1	0.0028

## Data Availability

Data collected during this study is available on request from the corresponding author.

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
