# Peer review of "Thyroxine and Thyroid-Stimulating Hormone in Own Mother’s Milk, Donor Milk, and Infant Formula"

_life, 2022, doi:10.3390/life12040584_

Round 1
Reviewer 1 Report
This study investigated the levels of albumin and TSH and thyroxine hormones in the breast milk of women with preterm and full-term babies during lactation, in pasteurized donor milk and also in infant formulas. The results show decreases in the concentrations of TSH and thyroxine in breastmilk of preterm, a drastic reduction of these hormones (and albumin) in milk having undergone Holder pasteurization and also an absence of these hormones in infant formulas.
Comments:
- Did the authors verify the linearity of their assays?
- Is there any point in testing directly skimmed milk? Have the assays been tested upstream on whole milk (not defatted)? Does this change the TSH and thyroxine assay values? Is this already known in the literature?
- What can explain the changes observed in breastmilk concentrations of TSH and thyroxine in preterm milk across lactation?
- Knowing that the concentrations of milk hormones change in the 6 months in preterms, why did the authors choose to show the values of average concentrations over the 6 months (table 3)?
- Can you please compare your values and those described in the discussion? Are your values comparable to other publications? If not, what could explain these variations across studies?
- A large paragraph of the discussion is focused on T3 and T4. Can you please explain why you choose to focus on these hormones in the discussion and not on TSH and thyroxine? Please make a link between T3, T4, TSH and thyroxine in the discussion as otherwise we do not understand why a large paragraph of the discussion is related to T3 and T4.
- The abstract may be edited to make it easier to read and understand.
- Some grammatical errors can be found across the text.
Author Response
Dear Reviewer 1:
On behalf of the coauthors of manuscript life-1624273, I thank you for your careful review of our manuscript. We have addressed each of the points raised by you below in italics and have made corresponding edits, where appropriate, using tracked changes in the manuscript.
Answers for Reviewer 1:
- Did the authors verify the linearity of their assays?
We used quality controls in known concentration every time, during the first measurement we added quality control sample to the tested milk samples to test and verify the method. The recovery rate was 98.6% in average.
- Is there any point in testing directly skimmed milk? Have the assays been tested upstream on whole milk (not defatted)? Does this change the TSH and thyroxine assay values? Is this already known in the literature?
Our current protocol,- based on other authors (Lemas et al, 2016; Aparicio et al, 2018) and our (Vass et al, 2020a; Vass et al, 2020b) previous works- were designed to measure the aqueous phase of breast milk. We have mentioned the separation technique as a limitation of our study, and we plan to measure whole breast milk samples in the future. TSH is a glycoprotein hormone, while thyroxine is a protein hormone presenting typically in the aqueous phase of breast milk, therefore we hypothyze that the rat of the possible loss is low. We have no information about the TSH or thyroxine content of the fat layer, however in one of our previous projects we analyzed the presence of different cytokines (CD40, Flt-3L), chemokines (MCP-1, RANTES, GRO, MIP-1ß, MDC, eotaxin, fractalkine), and epidermal growth factor (EGF) in the lipid layer of human milk (Vass et al, 2019).
Aparicio VA, Ocón O, Diaz-Castro J, Acosta-Manzano P, Coll-Risco I, Borges-Cósic M, Romero-Gallardo L, Moreno-Fernández J, Ochoa-Herrera JJ. Influence of a concurrent exercise training program during pregnancy on colostrum and mature human milk inflammatory markers: findings from the GESTAFIT Project. J Hum Lact. 2018 Nov;34(4):789-798.
Lemas DJ, Young BE, Baker PR 2nd, Tomczik AC, Soderborg TK, Hernandez TL, de la Houssaye BA, Robertson CE, Rudolph MC, Ir D, Patinkin ZW, Krebs NF, Santorico SA, Weir T, Barbour LA, Frank DN, Friedman JE. Alterations in human milk leptin and insulin are associated with early changes in the infant intestinal microbiome. Am J Clin Nutr. 2016 May;103(5):1291-300.
Vass RA, Kemeny A, Dergez T, Ertl T, Reglodi D, Jungling A, Tamas A. Distribution of bioactive factors in human milk samples. Int Breastfeed J. 2019 Feb 11;14:9. doi: 10.1186/s13006-019-0203-3.
Vass RA, Bell EF, Colaizy TT, Schmelzel ML, Johnson KJ, Walker JR, Ertl T, Roghair RD. Hormone levels in preterm and donor human milk before and after Holder pasteurization. Pediatr Res. 2020 Oct;88(4):612-617.
Vass RA, Roghair RD, Bell EF, Colaizy TT, Johnson KJ, Schmelzel ML, Walker JR, Ertl T. Pituitary Glycoprotein Hormones in Human Milk before and after Pasteurization or Refrigeration. Nutrients. 2020 Mar 4;12(3):687.
- What can explain the changes observed in breastmilk concentrations of TSH and thyroxine in preterm milk across lactation?
We hypothesize that after the suddenly interrupted pregnancy, maternal thyroid hormones may decrease slowly, which drop is present in the composition of breast milk (line 236-238).
- Knowing that the concentrations of milk hormones change in the 6 months in preterms, why did the authors choose to show the values of average concentrations over the 6 months (table 3)?
We did not detect significant monthly concentration changes in breast milk produced for term infants, therefore we decided to show average values to present the differences between term and preterm breast milk.
- Can you please compare your values and those described in the discussion? Are your values comparable to other publications? If not, what could explain these variations across studies?
Between line 259 and 269 we mentioned different results, since other works used different method, mainly radioimmunoassay, we did not compare directly our results, instead we were focusing on to calculate the average daily thyroxine hormone intake of a newborn.
- A large paragraph of the discussion is focused on T3 and T4. Can you please explain why you choose to focus on these hormones in the discussion and not on TSH and thyroxine? Please make a link between T3, T4, TSH and thyroxine in the discussion as otherwise we do not understand why a large paragraph of the discussion is related to T3 and T4.
Thank you for this advice, we have reconstructed this paragraph based on this absolutely reasonable suggestion, since the aim of our work was to focus on TSH and thyroxine.
- The abstract may be edited to make it easier to read and understand.
Thank you for the suggestion, we have made changes in the abstract.
- Some grammatical errors can be found across the text.
Thank you, we have checked the manuscript again.

Reviewer 2 Report
The manuscript by Vass et al. presents a study where human milk samples and infant formula samples are analyzed for thyroid stimulating hormone (TSH), thyroxine, and albumin. Human milk samples from both term and preterm mothers are included. To the best of my knowledge, few studies have reported data on human milk content of TSH, and in this sense the study is novel. In general, the manuscript is well written. However, some details around experimental design and sampling must be made more clear. Below some specific comments:
Sampling: The text states that “sample collection started during the 4th postpartum week and continued monthly, until six months” (lines 89-90). However, from the results reported, it is not evident that data were obtained for different time points up to 6 months. The manuscript should clearly provide information about sampling points for data reported. A figure illustrating experimental design and sampling times is recommended.
Discussion:
“When comparing plasma thyroid concentration of breast milk‐fed and formula‐fed infants no differences were found. The amount of T4 present in breast milk and formula is too low to influence the hypothyroxinemic state of preterm neonates” (Lines 234-237). It should be made more clear that this text is referring to a previous study and that it is not data obtained in the present study. Furthermore, the content of this text is actually quite pivotal and should be emphasized much more when discussing the results and implications of the present findings. Perhaps content of TSH in human milk is not important for hypothyroxinemic state of the newborn?
It should also be made clear that a limitation of the present study was that hormones were not measured in the blood of the infants, thus the impact of human milk content remains unknown.
In general the discussion is rather long and should be condensed.
Author Response
Dear Reviewer 2,
On behalf of the coauthors of manuscript life-1624273, I thank you for your careful review of our manuscript. We have addressed each of the points raised by you below in italics and have made corresponding edits, where appropriate, using tracked changes in the manuscript.
Answers to Reviewer 2
The manuscript by Vass et al. presents a study where human milk samples and infant formula samples are analyzed for thyroid stimulating hormone (TSH), thyroxine, and albumin. Human milk samples from both term and preterm mothers are included. To the best of my knowledge, few studies have reported data on human milk content of TSH, and in this sense the study is novel. In general, the manuscript is well written. However, some details around experimental design and sampling must be made more clear. Below some specific comments:
Sampling: The text states that “sample collection started during the 4th postpartum week and continued monthly, until six months” (lines 89-90). However, from the results reported, it is not evident that data were obtained for different time points up to 6 months. The manuscript should clearly provide information about sampling points for data reported. A figure illustrating experimental design and sampling times is recommended.
We thank the suggestion, a figure about experimental design was added to clear this issue.
Discussion:
“When comparing plasma thyroid concentration of breast milk‐fed and formula‐fed infants no differences were found. The amount of T4 present in breast milk and formula is too low to influence the hypothyroxinemic state of preterm neonates” (Lines 234-237). It should be made more clear that this text is referring to a previous study and that it is not data obtained in the present study. Furthermore, the content of this text is actually quite pivotal and should be emphasized much more when discussing the results and implications of the present findings. Perhaps content of TSH in human milk is not important for hypothyroxinemic state of the newborn?
Thank you, we have changed this sentence to make it clear. We hypothesize that TSH levels positively modulate thyroid stimulating hormone receptor in normal cells up to a certain limit, while down regulating TSHR at high concentrations (Akamizu et al, 1990).
Akamizu T, Ikuyama S, Saji M, Kosugi S, Kozak C, McBride OW, Kohn LD. Cloning, chromosomal assignment, and regulation of the rat thyrotropin receptor: expression of the gene is regulated by thyrotropin, agents that increase cAMP levels, and thyroid autoantibodies. Proc Natl Acad Sci U S A. 1990;87:5677–5681.
It should also be made clear that a limitation of the present study was that hormones were not measured in the blood of the infants, thus the impact of human milk content remains unknown.
Thank you, this sentence was added to the limitations (line 322-323).
In general the discussion is rather long and should be condensed.
We thank this suggestion and have made changes in the discussion.
Thank you for considering our manuscript for publication, we hope that we were able to improve our work based on the recommendations .
Yours sincerely,
Reka A. Vass MD PhD

Round 2
Reviewer 2 Report
I still find that the sampling procedures, especially timing of sampling, is unclear and that descriptions should be improved.
Another minor issue: How is preterm defined?
Author Response
Dear Reviewer 2:
On behalf of the coauthors of manuscript life-1624273, I thank you for accepting our answers. We have tried to improve the Figure, please find it attached and we made additional corrections in the text. Thank you for your careful review of our manuscript. We did not define “preterm”, since we have kept the original definition, newborns born before the 37th week of gestation.
Yours sincerely,
Reka A. Vass MD PhD
